# Transcriptome Analysis Identifies the Crosstalk between Dendritic and Natural Killer Cells in Human Cutaneous Leishmaniasis

**DOI:** 10.3390/microorganisms11081937

**Published:** 2023-07-29

**Authors:** Sara Nunes, Rafael Tibúrcio, Icaro Bonyek-Silva, Pablo Rafael Oliveira, Ricardo Khouri, Viviane Boaventura, Aldina Barral, Cláudia Brodskyn, Natalia Machado Tavares

**Affiliations:** 1Laboratory of Parasite-Host Interaction and Epidemiology (LaIPHE), Gonçalo Moniz Institute, Oswaldo Cruz Foundation (FIOCRUZ), Salvador 40296-710, Bahia, Brazil; sara_nunes2@hotmail.com (S.N.); rafaeltiburciops@gmail.com (R.T.); claudia.brodskyn@fiocruz.br (C.B.); 2Baiano Federal Institute (IFBaiano), Xique-Xique 47400-000, Bahia, Brazil; icaro.bonyek@gmail.com; 3Biology Institute (IBIO), Federal University of Bahia (UFBA), Salvador 40170-115, Bahia, Brazil; pablorafael_ssa@hotmail.com (P.R.O.); ricardo.khouri@fiocruz.br (R.K.); viviane.boaventura@fiocruz.br (V.B.); abarral@fiocruz.br (A.B.); 4Laboratory of Infectious Diseases Transmitted by Vectors (LEITV), Gonçalo Moniz Institute, Oswaldo Cruz Foundation (FIOCRUZ), Salvador 40296-710, Bahia, Brazil

**Keywords:** *Leishmania braziliensis*, natural killer cells, dendritic cells, transcriptome, cutaneous leishmaniasis

## Abstract

Skin ulcers of cutaneous leishmaniasis (CL) are characterized by a localized inflammatory response mediated by innate and adaptive immune cells, including dendritic cells (DC) and natural killer (NK) cells. Bidirectional interactions between DCs and NK cells contribute to tailor leishmaniasis outcome. Despite advances in the *Leishmania* biology field in recent decades, the mechanisms involved in DC/NK-mediated control of *Leishmania* sp. pathogenesis as well as the cellular and molecular players involved in such interaction remain unclear. The present study sought to investigate canonical pathways associated with CL arising from *Leishmania braziliensis* infection. Initially, two publicly available microarray datasets of skin biopsies from active CL lesions were analyzed, and five pathways were identified using differentially expressed genes. The “Crosstalk between DCs and NK cells” pathway was notable due to a high number of modulated genes. The molecules significantly involved in this pathway were identified, and our findings were validated in newly obtained CL biopsies. We found increased expression of TLR4, TNFRSF1B, IL-15, IL-6, CD40, CCR7, TNF and IFNG, confirming the analysis of publicly available datasets. These findings reveal the “crosstalk between DCs and NK cells” as a potential pathway to be further explored in the pathogenesis of CL, especially the expression of CCR7, which is correlated with lesion development.

## 1. Introduction

Leishmaniasis is a chronic inflammatory disease caused by protozoan parasites of the genus *Leishmania* whose main vectors are phlebotomine sandflies. This disease is considered a worldwide public health problem, as it is present in 102 countries, placing approximately 1 billion people at risk of infection, with an estimated 12 million people currently infected. Each year, approximately two million new cases are registered, resulting in 70,000 deaths [1,2,3,4,5].

Cutaneous leishmaniasis (CL) is the most common form of human leishmaniasis, accounting for approximately 95% of cases, on average [3]. In Brazil, *Leishmania (Viannia) braziliensis* [6] is the major etiological agent of CL, which is characterized by ulcerated skin lesions, i.e., small papules formed in the initial period following parasite inoculation. During disease progression, ulcerated lesions, whose pathological hallmarks are a necrotic center and elevated borders, replace these papules [7]. It has been well documented that the adaptive branch of immunity is crucial to the establishment of a protective response against Leishmania. In the context of *L. braziliensis* infection, the Th1 response is fundamental for parasite elimination and lesion resolution; however, this same immune response promotes tissue damage [8,9,10,11]. High levels of tumor necrosis factor-α (TNF-α) and interferon-γ (IFN-γ) are detected in infected patients and may be associated with dendritic cells (DC) and natural killer (NK) cells [12,13]. Both are innate cellular components and important regulators of the adaptive immune response against infectious pathogens [8,11,12,13].

DCs are potent professional antigen-presenting cells (APCs) that effectively link innate and adaptive responses. Furthermore, DCs are a heterogeneous cell population and classified according to origin, phenotype, tissue distribution and function [11]. Under resting conditions, dendritic cells (DCs) are typically considered immature. However, upon antigen capture, they undergo a maturation process that leads to their activation and the initiation of immune responses. This maturation process involves several changes within the DCs [13]. Firstly, upon antigen capture, DCs up-regulate the expression of major histocompatibility complex (MHC) molecules on their cell surface. MHC molecules are responsible for presenting antigens to T cells, enabling T cell recognition and activation. By increasing the expression of MHC molecules, DCs enhance their ability to present antigens to T cells effectively [13]. The increased expression of co-stimulatory molecules, such as CD80, CD86 and CD40, is also notably important to this process. Moreover, elevated CCR7 expression has been correlated with cell migration [13,14,15,16].

NK cells constitute a subset of granular lymphocytes that play a dual role in immunity. These cells can orchestrate cytotoxic or immunomodulatory processes mainly involved in the recognition and elimination of stressed cells from different types: infection, tumor-related transformations, and physical damage [12,16,17]. This recognition occurs through a variety of mechanisms, including physical contact via either activation or inhibition receptors, or both. In addition, NK cells can become activated through the release of cytokines, such as IL-2, IL-12, IL-15 and IL-18 [18,19,20]. When stimulated, these cells secrete a distinct set of cytokines, including IFN-γ (the major cytokine released by NK cells), TNF-α and granulocyte macrophage colony stimulating factor (GM-CSF), as well as chemokines, such as CCL2, CCL3, CCL4 and CCL5 [18,21,22,23].

Abundant evidence suggests that NK cells regulate the functions of other cells via the production of cytokines and chemokines and/or direct cell–cell contact (receptor–ligand interactions) [19,24]. This immunological axis has been extensively described by the crosstalk between NK cells and DCs [19,24]. Several studies suggest that NK–DC interactions can accelerate DC maturation via NKp30 receptor [13], as well as its migration and activation. During viral infections, NK cells can induce the death of infected DCs [25,26,27]. In turn, DCs also play an important role in the survival, proliferation, activation and cytotoxicity of NK cells, mainly through the release of pro-inflammatory cytokines and lipid mediators [11,13,26].

The present study sought to analyze transcriptome datasets to identify canonical pathways associated with the innate immune response in cutaneous leishmaniasis (CL) lesions caused by *Leishmania braziliensis*. The investigation of this pathway in the context of parasite infections has been previously unclear, and the study aims to shed light on this aspect, providing new insights and potential strategies for the management of this infection. By examining the transcriptome profiles, the study aims to uncover specific molecular pathways and mechanisms involved in CL, which could be targeted for the development of novel therapeutic interventions. The findings from this research have the potential to broaden our understanding of the host-parasite interaction in CL and pave the way for innovative approaches to tackle this challenging infectious disease.

## 2. Materials and Methods

### 2.1. Microarray Datasets

Transcriptomic data sets were obtained from the Gene Expression Omnibus (GEO) database (www.ncbi.nlm.nih.gov/geo/ (accessed on 15 February 2022) using the following search terms: ‘human’, ‘biopsy’, ‘skin’ and ‘*Leishmania braziliensis*’. The dataset published by Novais et al. (GSE55664) [28] was generated using Illumina HT12v4 platform (GPL10558) from the analysis of 10 healthy control (HC) skin samples and 25 lesion biopsies from patients with CL from Corte de Pedra, Bahia (Brazil). Another dataset published by Oliveira et al. (GSE63931) [29] was generated using Agilent SurePrint GE Human G3v2 platform (GPL17077). This dataset compared eight healthy control skin samples and eight lesion biopsies from patients with CL in Itabuna, Bahia (Brazil). A total of 47,304 probes and 22,965 transcripts were detected by the Illumina platform, while the Agilent platform presented 50,561 probes and 24,079 transcripts (Appendix A). The resulting Venn diagram illustrates that nearly 21,000 transcripts were common to these datasets, which were subsequently used for differential gene expression analysis.

### 2.2. Identification of Differential Gene Expression

Raw gene expression data were analyzed using the Multi Experiment Viewer (mev.tm4.org/ (accessed on 15 March 2022)) microarray data analysis tool to generate heat maps representing significantly DEGs according to the following criteria: a previously evaluated FDR adjusted *p*-value < 0.05, and fold change of ≤−2 or ≥+2. The default parameters of MEV were used. The hierarchical clustering considered was the ‘Gene tree’ and ‘Sample tree’ and the current metric by ‘Euclidean distance’. The normalization was performed by ‘gene’ and ‘sample’ and the set color scale limits according to the displayed default value.

### 2.3. Canonical Pathways Analysis

Each dataset (GSE55664 and GSE63931) was analyzed by Ingenuity Pathway Analysis (IPA) (www.qiagenbioinformatics.com (accessed on 15 March 2022) to identify pathways related to CL. The search term “Immune response” was applied to identify specific pathways of interest, considering only DEGs in each dataset.

### 2.4. Validation Cohort and Biopsy Obtainment

A cohort of eight patients was enrolled to validate the DEGs obtained from the analysis of public datasets. These patients were evaluated at the municipal public health clinic in Jiquiriçá, an endemic area for *L. braziliensis* in Bahia, Brazil. The criteria for diagnosis were clinical symptoms of CL, histopathological analysis to confirm CL and a positive delayed-type hypersensitivity response to *L. braziliensis* antigens. In addition, qPCR was used to confirm the parasite species. Skin biopsies were collected at the border of the lesion using a 4 mm diameter punch prior to therapy. Healthy skin samples were obtained from six volunteers who underwent elective plastic surgery and resided in a non-endemic area with no history of CL. Written informed consent was obtained from all study subjects prior to biopsy collection procedures.

### 2.5. RNA Extraction 

Total RNA was isolated from skin samples of patients and controls. The samples were macerated and vortexed in 700 μL of TRIzol reagent (Life Technologies, Carlsbad, CA, USA). After, 140 μL of chloroform (Sigma-Aldrich, St. Louis, MO, USA) was added and vigorously vortexed followed by incubation for 5 min at room temperature. The aqueous phase was recovered, mixed with 500 μL 70% ethanol, and RNA was obtained using the RNeasy mini kit (Qiagen, Hilden, Germany) in accordance with the manufacturer’s instructions. RNA yield was assessed using a NanoDrop1000 spectrophotometer (Thermo Fisher Scientific, Waltham, MA, USA).

### 2.6. Gene Selection and Quantitative Real-Time PCR

To validate the results of our DEG analysis, eight genes were selected for relative quantification considering their fundamental importance to the pathway “Crosstalk between DC and NK cells” identified by IPA: Surface receptors TLR4, CD40 and CCR7; Cytokines produced by DCs TNF, IL15, IL6; TNFRSF1B, responsible for recognizing TNF; and IFNG, the cytokine produced by NK cells.

A total of 1 μg of RNA was converted into complementary DNA (cDNA) using a SuperScript™ III Reverse Transcriptase kit (Invitrogen Cat: 18080085). Quantitative real-time PCR was performed using ABI7500 Real-Time PCR system (Applied-Biosystems, Waltham, MA, USA) with 10 uL of SYBRGreen Real-Time PCR Master Mix. All relevant primers were synthesized by Integrated DNA Technologies. The relative expression of these genes was evaluated using the 2^−ΔΔCt^ method, and results were normalized using the ACTB housekeeping gene.

### 2.7. Ethics Statement

This study was conducted in accordance with the principles of the Declaration of Helsinki and local ethical guidelines. This study was approved by the Ethics Committee of Instituto Gonçalo Moniz (Salvador, Bahia, Brazil—(protocol number CAAE 47120215.8.0000.0040). All patients who participated in the research provided written consent for sample collection (TCLE—Free and Informed Consent Form) and subsequent analysis.

### 2.8. Statistical Analysis

GraphPad Prism 8 (GraphPad Software, San Diego, CA, USA) was used for statis-tical evaluation and *p*-values < 0.05 were considered significant. For comparisons between groups, the Shapiro–Wilk test was used to detect data normality. For the evaluation of 2 groups, both with a normal distribution, Student’s *t*-test was applied, and for samples with non-normal distribution, the Mann–Whitney test was used. In analyzes with more than 2 groups, we used the analysis of variance test (ANOVA) for samples with normal distribution and Kruskal–Wallis followed by Dunn’s post-test for samples with non-normal distribution. Graphs were presented with a 95% confidence interval for the mean (normal distribution) or median with interquartile deviation (non-normal distribution). The results were represented with the median values obtained in each condition. Correlation analyses were performed by Spearman test and the matrix summarizes the statistically significant correlations. The correlation matrix is reordered according to the correlation coefficient using “hclust” method (http://www.sthda.com/english/rsthda/correlation-matrix.php (accessed on 20 May 2022). The 2^−^^ΔΔCt^ method is a widely used approach in quantitative real-time PCR (qPCR) data analysis. It is commonly used to determine the relative gene expression levels between different samples or experimental conditions.

## 3. Results

### 3.1. Specific Gene Expression Profile in Biopsies of Human CL

We first assessed raw expression data from public transcriptomes of active CL lesions obtained from different endemic areas. A total of 2368 genes were found to be differentially expressed (DEG: *p* < 0.05 and ≤−2 FC ≥ +2) in the GSE55664 dataset: 1243 were up-regulated (yellow) and 1125 were down-regulated (blue). In the GSE63931 dataset, 6152 DEGs were identified: 2851 up-regulated (yellow) and 3301 down-regulated (blue) (Figure 1A). The resulting Venn diagram details all DEGs (Figure 1B) and illustrates that 715 transcripts were similarly down-regulated (blue) and 1022 were up-regulated (yellow) between both datasets. In all, a significant number of DEGs were found to be present among these two different cohorts, suggesting a specific gene expression profile in CL.

### 3.2. Canonical Pathway “Crosstalk between Dendritic Cells and Natural Killer Cells” Is Significantly Induced in CL Caused by L. braziliensis

Next, IPA analysis were used to identify canonical pathways related to CL in both datasets, GSE55664 (Figure 1C) and GSE63931 (Figure 1D), according to previously found DEGs (Figure 1A,B). Based on the resulting *p* values, a high degree of similarity was observed between pathways related to immune response in each dataset: Communication between Innate and Adaptive Immune Response; Crosstalk between Dendritic Cells and Natural Killer Cells; Granulocyte Adhesion and Diapedesis and Interferon Signaling (Figure 1C,D). Additional investigations were carried out on other inflammatory skin diseases, such as psoriasis (Figure 1E) and lupus (Figure 1F). Despite these efforts, the pathway profile seems to be specific for CL.

In GSE55664 dataset (Figure 1G), approximately 65% of the transcripts were up-regulated (yellow), while only 3% were down-regulated (blue), in the pathway ‘Communication between Innate and Adaptive Immune Response’ (Appendix A). Considering the GSE63931 dataset (Figure 1G), 75% of the transcripts were up-regulated and 6% were down-regulated in this pathway (Appendix A). Similarly, the pathway ‘Crosstalk between Dendritic Cells and Natural Killer Cells’ presented 60% up-regulated and 3% down-regulated transcripts in the GSE55664 dataset (Figure 1H), while 74% of these were up-regulated and 6% were down-regulated in the GSE63931 dataset (Figure 1H). These results indicate that both canonical pathways may play a significant role in human CL due to the absolute numbers of up-regulated transcripts found by two independent studies.

Since Innate and Adaptive Immune Response have been extensively studied in CL, this study will focus on the second canonical pathway identified, Crosstalk between Dendritic and Natural Killer Cells, which has been poorly explored in the context of leishmaniasis.

### 3.3. DEGs from “Crosstalk between DCs and NK Cells” Pathway Are Able to Distinguish between Healthy Skin and CL Lesions

The expression levels of each DEG in the ‘Crosstalk between Dendritic Cells and Natural Killer Cells’ pathway was assessed to determine its relevance in CL. We found 22 and 28 differentially expressed genes (*p* < 0.05; fold change ≥2 of increased or decreased expression) in the GSE55664 (Figure 2A) and GSE63931 (Figure 2B) datasets, respectively (Appendix A). Opposing DEG expression profiles were revealed when comparing between CL patients and health controls (blue and yellow colors). Subsequent hierarchical clustering analysis indicated that the gene expression profile of ‘Crosstalk between Dendritic Cells and Natural Killer Cells’ pathway differentiated CL patients from controls (Figure 2A,B), confirming the opposite expression of genes between CL lesions and healthy skin. Of the 22 DEGs in GSE55664 (Figure 2C), 21 were up-regulated (yellow), while only one was down-regulated (blue). Of the twenty-eight DEGs identified in GSE63931, twenty-six were up-regulated, while only two were down-regulated (Figure 2D), suggesting that ‘Crosstalk between Dendritic Cells and Natural Killer Cells’ pathway is highly activated during CL. Taken together, the results from both datasets demonstrate that CL lesions and healthy control skin have distinct gene expression profiles, especially regarding the pathway ‘Crosstalk between Dendritic Cells and Natural Killer Cells’, whose role in CL has been neglected.

### 3.4. IFNG and CCR7 Genes Overlap between Datasets and Are the Most Modulated Genes of the “Crosstalk between DCs and NK Cells” Pathway

To explore DEGs common to both datasets, we constructed a Venn diagram to identify overlapping genes. Figure 3A shows that 20 genes overlap between both datasets according to the larger to smaller fold change: IFNG, FASLG, CCR7, CD80, IL2RG, CSF2, TLR7, IL2RG, TNF, CD83, IL6, TNFRSF1B, TLR4, IL15, CD40, HLA-E, CD209, IL15RA, TNFSF10 and FAS (Figure 3A). To validate this finding, RT-qPCR was performed to detect eight genes of the ‘Crosstalk between DCs and NK cells’ pathway. New samples of human CL lesions (*n* = 8) were obtained from a different endemic area, as well as healthy controls (*n* = 7) from a non-endemic area. The relative expression of TLR4 (Figure 3C), IL6 (Figure 3D), IL15 (Figure 3E), CD40 (Figure 3F), CCR7 (Figure 3G), TNF (Figure 3H), TNFRSF1B (Figure 3I) and IFNG (Figure 3J) was found up-regulated by more than two-fold compared to controls, confirming the induction of ‘crosstalk between DCs and NK cells’ pathway in CL. The values of fold change for each validated gene are summarized in Figure 3B, illustrating distinct areas occupied by healthy and CL groups (Figure 3B). Remarkably, the expression of IFNG and CCR7 exhibited substantial up modulation in CL samples (Figure 3B), suggesting that ‘crosstalk between DCs and NK cells’ pathway may play a role in the immune response against *L. braziliensis*.

### 3.5. The Expression of CCR7 Negatively Correlates with the Lesion Size

Considering that gene expression could be associated with the lesion stage (Figure 4), a correlation matrix was constructed with the validated genes (Figure 3) and the lesion size. Only statistically significant correlations are shown in the matrix (Figure 4A), revealing a positive association between IL15 and TLR4, as well as CCR7 (Figure 4B). The lesion size negatively correlates with CCR7, and we also found a positive correlation of TNF/TNFRSF1B with CCR7, IL6 and IFNG (Figure 4C), suggesting that TNF/TNFRSF1B signaling orchestrate the inflammation mediated by CCR7, IL6 and IFNG, nourishing an inflammatory loop that can lead to ulceration and a chronic lesion.

Together, these findings reveal that the crosstalk between DCs and NK cells may play a significant role in the ulceration of CL lesions and maintenance of inflammation in the site of infection. However, in vitro assays with DCs and NK cells are necessary to confirm the consequences of this crosstalk for the control of *Leishmania* infection and also for inflammation. In sum, we found that this underappreciated pathway is up-regulated in CL lesions, which may maintain inflammation in the site of infection and influence lesion development.

## 4. Discussion

Crosstalk between DCs and NK cells plays a critical role in the regulation of tumor growth, bacterial infection, and antiviral response, leading to an efficient immune response mediated by soluble factors and interaction between these two cells. However, the mechanisms involved in parasite infection as well as the precise nature of the interactions occurring between these cells remain unclear [25,30,31,32,33].

The present study identified some canonical pathways related to ‘innate immune response’ involved in human CL. Interestingly, we found that “Crosstalk between DCs and NK cells” was one of the most significant pathways associated with active lesions caused by *L. braziliensis* [34]. Our results provide evidence that bidirectional interactions between DCs and NK cells are highly involved in the innate and adaptive immune response to CL in patients from different endemic areas. Transcriptome analysis revealed that molecules identified in this pathway are significantly modulated in both datasets [33,34]. In addition, these findings were validated in other biopsies from CL patients and then compared with healthy samples by real-time PCR. The present results are in agreement with other studies that have shown the ‘Crosstalk between DCs and NK cells’ pathway regulated in other inflammatory diseases. Niu and collaborators analyzed pathways involved in rheumatoid arthritis and type 2 diabetes to identify shared pathways. This represents a new strategy to discover novel treatment targets for these diseases [35].

Several studies have demonstrated the importance of this pathway in viral infections. A study employing systems biology analysis identified genes and canonical pathways involved in tissue tropism in West Nile Virus (WNV) infection. Comparisons of gene expression among different tissues in WNV infection in mice identified the ‘Crosstalk between Dendritic Cells and Natural Killer Cells’ as important in this infection, suggesting a new antiviral therapy against flavivirus infections [36]. Another study showed how murine cytomegalovirus (MCMV) infection can modulate the expression of interferon by DC and NK cells [37]. On the other hand, NK cells have been largely studied in cancer since its cytotoxic function may be useful to the development of anti-cancer therapies. The gene expression analysis of ex vivo expanded and freshly isolated NK cells from cancer patients identified the up-regulation of “crosstalk between DCs and NK cells” pathway, suggesting that interaction between these cells participates in the cellular immune response [38,39]. In addition, a previous study identified molecular signatures mostly associated with this crosstalk, which have been predictive of relapse-free survival in breast cancer patients. These authors proposed a prognostic signature that highlights the cooperation between the innate and adaptive immune components within the tumor microenvironment [40]. Other studies have also shown that interactions between these cells can influence the mechanism involved in autoimmune diseases. Rai and collaborators found ‘crosstalk between DCs and NK cells’ associated with uniquely expressed transcripts in distinct Systemic Lupus Erythematosus patient subsets [41]. Taken together, these findings highlight the role of these cells in modulating innate and inflammatory adaptive responses.

Although the role of crosstalk between DCs and NK cells in leishmaniasis remains poorly understood, many studies have demonstrated the importance of DCs and NK cells in *Leishmania* infection. Studies have shown that *L. major*, *L. infantum* and *L. amazonensis* exhibit a mechanism of immune evasion through the inhibition of DC activation, migration and immune function, which interferes in the induction of adaptive immune responses [42,43,44,45]. Within the of innate immunity, NK cells occupy a crucial position due to their dual functionality in eradicating infected cells and production of cytokine that help the activation of other immune cells. Unraveling the precise contributions of NK cells to the pathogenesis and resolution of CL has been the subject of intensive investigation, aiming to provide comprehensive insights into the disease’s dynamics and therapeutic possibilities [46,47].

It is known that interactions between DCs and NK cells can result in the amplification of inflammatory activity by both cells [48]. In vitro studies with murine cells demonstrate that coculturing of *L. amazonensis*-infected DCs with resting NK cells results in increased activation of both cells, beyond augmented production of proinflammatory cytokines, including IFN-γ. In this same study, murine DCs infected by *L. braziliensis* amastigotes became activated, which was not the case in *L. amazonensis* infection [49]. Moreover, DCs were also involved in the control of *L. major* infection via crosstalk with NK cells, leading to their activation and release of IFN-γ [50]. Our results indicate that TLR4 was up-regulated in biopsies from patients with CL compared to healthy controls. This finding may be associated with the production of IL-15, which is essential to the survival of NK cells. Interactions between infected DCs and NK cells can contribute to the maturation/activation processes of DCs and result in their activation. This crosstalk results in the mutual production of TNF-α, as well as increased expression of the molecules involved in DC migration/activation (CCR7, CD40, CD80 and CD83) and the release of proinflammatory cytokines, such as IL-6. In addition, this crosstalk culminates in IFN-γ production by NK cells, leading to increased effector function and the activation of other leukocytes.

## Figures and Tables

**Figure 1 microorganisms-11-01937-f001:**
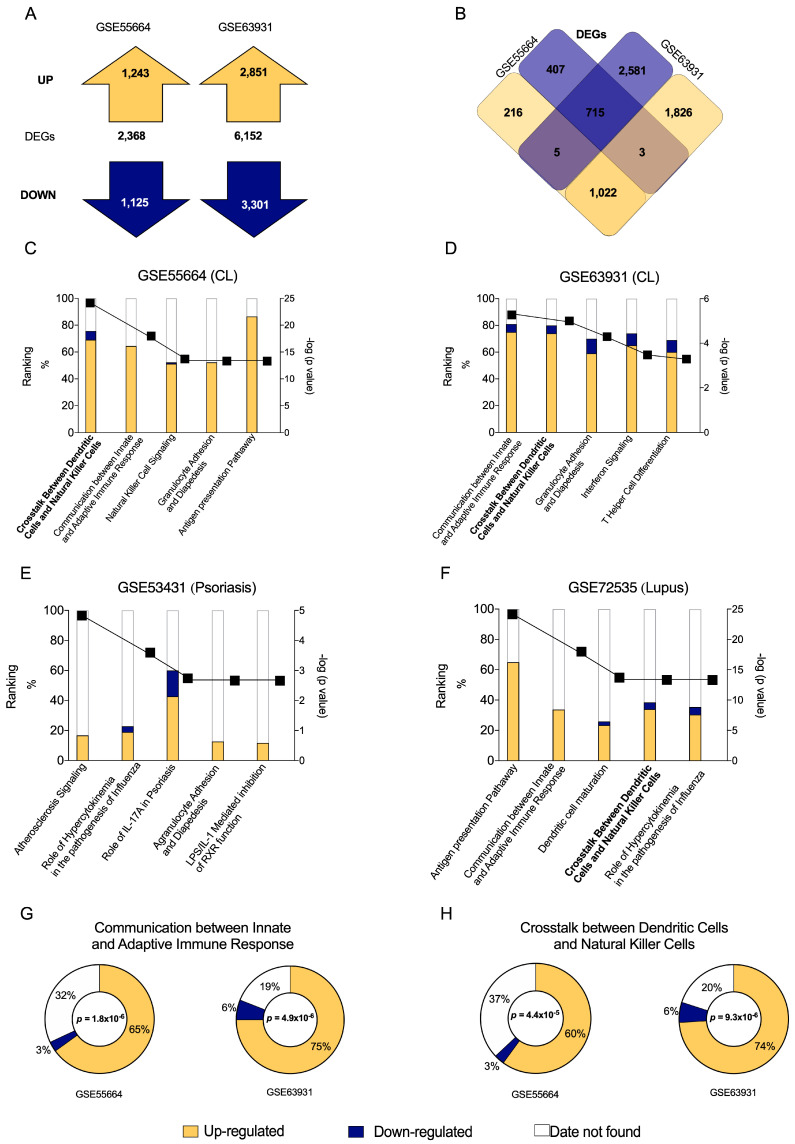
Differentially Expressed Genes (DEGs) in CL patient skin lesions obtained from two publicly available datasets and canonical pathways associated to disease. (**A**) Total number of DEGs, including up-regulated (yellow) and down-regulated (blue) genes, contained in the GSE55664 and GSE63931 datasets. (**B**) Venn diagram showing DEG overlap between the two datasets. Graphs illustrate the numbers of up-regulated (yellow) and down-regulated (blue) molecules, as well as the number of missing molecules (white) in data sets for CL GSE55664 (**C**) and GSE63931 (**D**). Additionally, up- and down-regulated molecules were evaluated for Psoriasis (**E**) and Lupus (**F**) to be able to compare the canonical pathways with CL. Of the five most significant canonical pathways related to immune response in both transcriptomes, the two most important pathways identified by Ingenuity Pathway Analysis were Communication between Innate and Adaptive Immune Response (**G**) and Crosstalk between Dendritic Cells and Natural Killer Cells (**H**).

**Figure 2 microorganisms-11-01937-f002:**
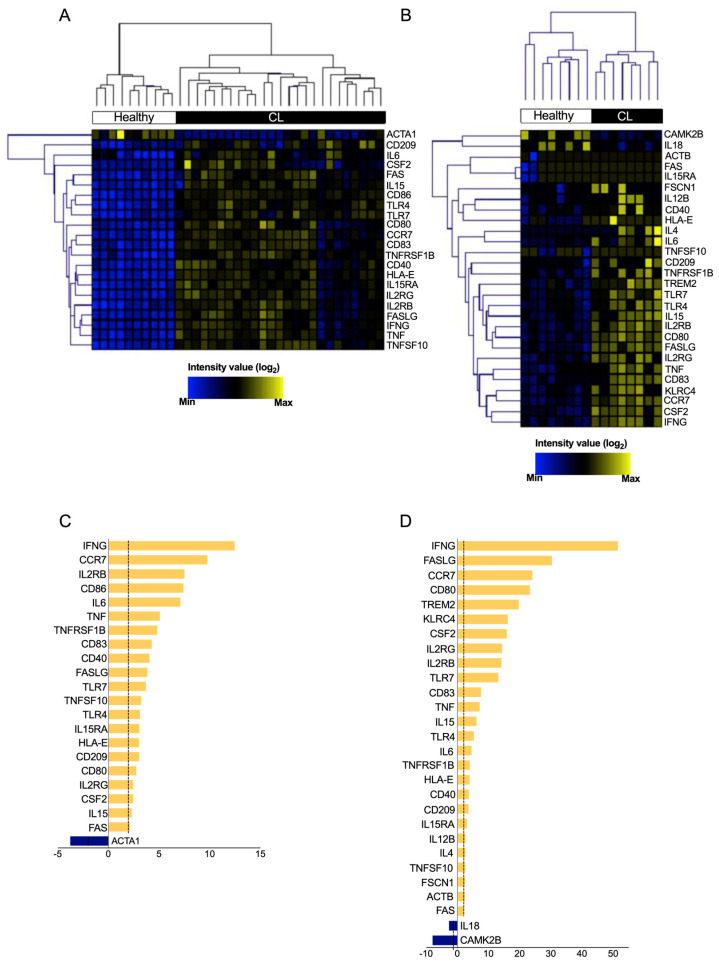
Hierarchical clustering of transcript expression profiles differentially expressed in crosstalk between DCs and NK Cells in cutaneous leishmaniasis. The expression profile of the significantly modulated transcripts that compose the canonical pathway ‘Crosstalk between Dendritic Cells and Natural Killer Cells’ (*p* < 0.05 and −2 < fold change > 2) comparing healthy control samples (white) and skin biopsy from patients with CL (black). Heat maps showed the 21 (**A**) and 28 (**B**) transcripts in GSE55664 and GSE63931, respectively. Furthermore, the graph of fold change for each transcriptome, GSE55664 (**C**) and GSE63931 (**D**).

**Figure 3 microorganisms-11-01937-f003:**
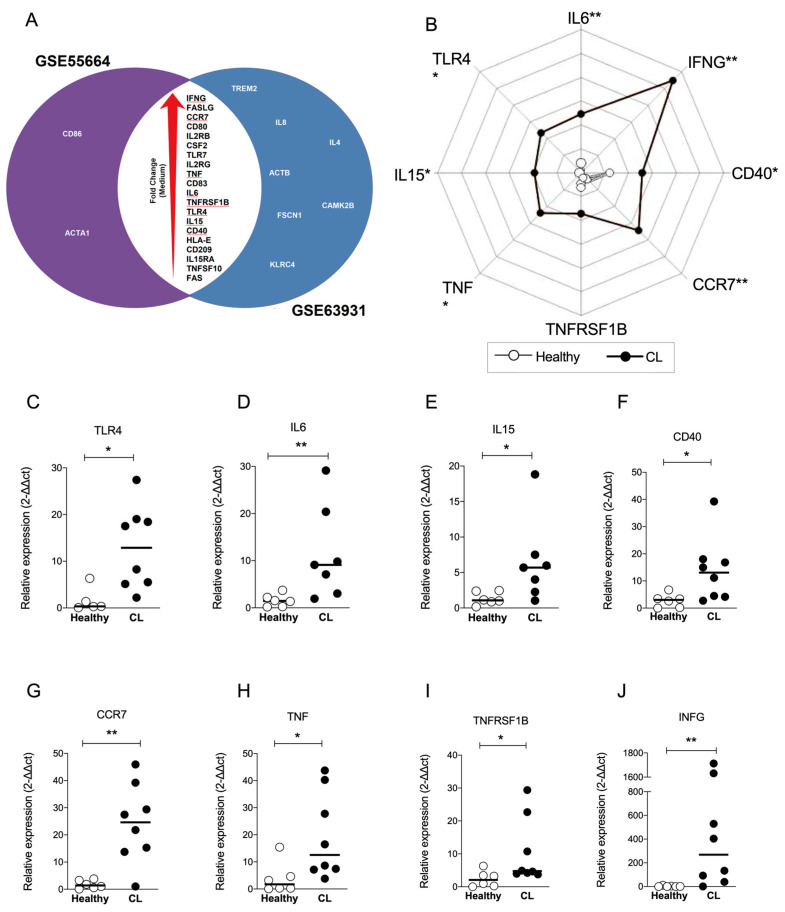
Identification and validation of overlap of transcripts significantly up-regulated in interaction between DCs and NK cells in CL. (**A**) Venn Diagrams showing the group of transcripts in common for each data sets (GSE55664 and GSE63931) to the pathway ‘Crosstalk Between Dendritic Cells and Natural Killer Cells’ in CL. The data are shown in increase order of fold change (red arrow). Genes highlighted by the red lines were further validated by q-PCR. (**B**) A representative profile of geometric mean values (log10 transformed) for indicated genes displayed for each group. Validation of the relative expression from biopsies from CL patients (*n* = 8) compared with healthy controls (*n* = 6) showed the variation of TLR4 (**C**), IL6 (**D**), IL15 (**E**), CD40 (**F**), CCR7 (**G**), TNF (**H**), TNFRSF1B (**I**) and IFNG (**J**). Data analyzed by Mann–Whitney t test comparing CL biopsies (black) with healthy control (white). Bars represent the median ± SEM, * *p* ≤ 0.05 and ** *p* ≤ 0.01.

**Figure 4 microorganisms-11-01937-f004:**
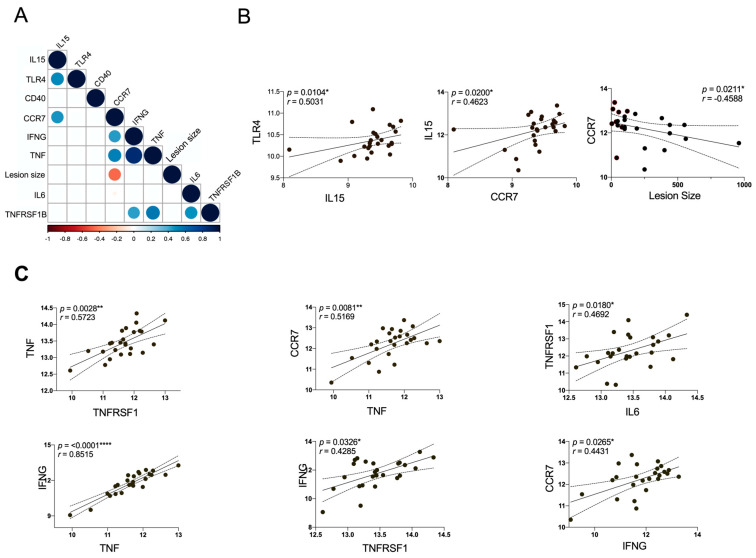
Correlation matrix for all DEGs associated to DC/NK pathway and skin lesion size from CL patients. (**A**) The correlation between DEGs and clinical data of patients is shown in the matrix graph, the circles and corresponding sizes represent *r*-values (*r* > 0.50) for each Spearman correlation. Colors represent the directionality of the correlation (blue infers positive correlation, whereas red indicates negative correlations). (**B**,**C**) The significant Spearman correlation between DEGs and lesion size are represented in linear regression analysis. Each symbol represents an individual CL patient compared with control. The *p*-values were calculated for correlation efficiency and were considered significant at * *p* ≤ 0.05, ** *p* ≤ 0.01 and **** *p* ≤ 0.0001.

## Data Availability

The data presented in this study are openly available in www.ncbi.nlm.nih.gov/geo/, at doi:10.1038/jid.2014.305 and doi:10.4049/jimmunol.1402047.

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
