# Peer review of "Transcriptome Analysis Identifies the Crosstalk between Dendritic and Natural Killer Cells in Human Cutaneous Leishmaniasis"

_microorganisms, 2023, doi:10.3390/microorganisms11081937_

Round 1

Reviewer 1 Report

The manuscript titled "Transcriptome analysis identifies the crosstalk between dendritic and natural killer cells in human cutaneous leishmaniasis" presents an investigation into the gene expression patterns associated with cutaneous leishmaniasis (CL) using microarray datasets. Overall, the study is well-conducted and provides valuable insights into the molecular mechanisms underlying CL. However, there are a few areas that require clarification and improvement before the manuscript can be considered for publication.

a. The authors should provide more details about the Microarray Datasets section (2.1). It is unclear how many datasets were obtained from the Gene Expression Omnibus (GEO) database and which specific datasets were used for the analysis. Please provide the accession numbers and a brief description of each dataset.

b. In the Identification of differential gene expression section (2.2), the authors mention the use of the Multi Experiment Viewer (MEV) tool for analyzing raw gene expression data. However, it would be helpful to describe the specific settings or parameters used in the analysis and provide a reference or a brief explanation of the tool for readers who may not be familiar with it.

c. The authors state that Ingenuity Pathway Analysis (IPA) was used for canonical pathways analysis (section 2.3). Please provide more information about the criteria used to identify significantly differentially expressed genes (DEGs) in each dataset. Additionally, clarify whether the search term "Immune response" was applied to both datasets or only to one of them.

d. In the Validation cohort and biopsy obtainment section (2.4), please provide more information on the characteristics of the enrolled patients, such as age, gender, and clinical presentation. Additionally, describe the methods used for histopathological analysis and delayed-type hypersensitivity response assessment.

e. In the Gene Selection and Quantitative Real-Time PCR section (2.6), it would be beneficial to explain why these eight specific genes were chosen for validation and provide references to support their fundamental importance to the pathway "Crosstalk between DC and NK cells" identified by IPA.

f. The authors mention the use of the 2ΔΔCt method for relative quantification of gene expression (section 2.6). Please provide a brief description or a reference to explain this method for readers who may not be familiar with it.

Ethics Statement:

a. In the Ethics Statement section (2.7), please provide more details about the procedures followed to obtain written informed consent from the study subjects and ensure their privacy and confidentiality.

Statistical Analysis:

a. The authors mention the use of GraphPad Prism 8 for statistical evaluation (section 2.8). Please specify which specific statistical tests were used and provide a reference or a brief explanation of the software for readers who may not be familiar with it.

b. It is unclear from the manuscript how the correlations were analyzed and what variables were included in the correlation matrix (section 2.8). Please provide more details on the statistical methods used for the correlation analysis and describe the variables included in the matrix.

Addressing these points will significantly improve the clarity and robustness of the manuscript. Overall, the study presents important findings on the differential gene expression in cutaneous leishmaniasis and has the potential to contribute to our understanding of the disease.

minor

Reviewer 2 Report

The study “Transcriptome analysis identifies the crosstalk between dendritic and natural killer cells in human cutaneous leishmaniasis” was addressed in an integrated and rigorous manner. The outcomes are of utmost importance in the context of Leishmania sp control. Generally, the writing and arguments in the paper are technically valid and the text is easy to read. This work constitutes a good contribution to the field. The title is relevant and informative and the abstract matches the whole text. It indicates the main topics, results, and the research question is for me clearly outlined.

·         The introduction is concise and good but needs more updated references.

·         Remove the paragraph from 29-35 (0. How to Use This Template) 

·         Add relevant and updated references to support lines from 38 to 40

·         This sentence must be supported by a reference: “Cutaneous leishmaniasis (CL) is the most common form of human leishmaniasis, accounting for approximately 95% of cases, on average.”

·         The same for this sentence: “During disease progression, ulcerated lesions, whose pathological hallmarks are a necrotic center and elevated borders, replace these papules” and for this: “High levels of tumor necrosis factor-α (TNF-α) and interferon-γ (IFN-γ) are detected in infected patients and may be associated with dendritic cells (DC) and natural killer (NK) cells”.

·         The sentence from line 58 to 62 is very long, please consider breaking it into small sentences.

·         Lines from 86 to 92 are results: the findings may be added in the abstract, Results-Discussion, and Conclusion sections, not in the Introduction section.

· At the end of the Introduction Section, the authors should clearly outline the hypothesis and the primary objectives of the study.

·         The figures are relevant and the results are well described

·         Ethics: The protocol of this study was approved by Ethics Committee.

·         Please highlight well the new knowledge generated.

·         Add a Limitations and Implication subsection of the study.

Reviewer 3 Report

Authors must remove topic 0 (zero) from the manuscript: "How to Use This Template"

Surely the authors forgot to delete this section that describes how to use the Journal MICROORGANISMS template

The manuscript is very well presented, very well written, the figures represent very well what is discussed in the manuscript. The research group has a lot of experience in this area of study and has already produced great contributions in the studies of leishmaniasis.

Overall, this manuscript analyzed transcriptome datasets to identify canonical pathways related to the innate immune response in CL lesions caused by Leishmania braziliensis. The authors describe that 'crosstalk between dendritic cells and NK cells' is one of the most important upregulated pathways in active CL lesions. The authors also elegantly demonstrate these results in freshly obtained skin biopsies. Among validated molecules, CCR7 negatively correlates with CL lesion size. Thus revealing a potential role for cell migration in lesion ulceration. Summarizing what was demonstrated in the MS, the authors describe that the results obtained place the interaction between DCs and NK cells as an important axis for the immunopathogenesis of CL.

This revelation is interesting, as this route has always been overlooked by different study groups. however, thanks to studies like this one, new ways are being discussed in an attempt to understand the biology of cutaneous leishmaniasis.

Reviewer 4 Report

From two transcriptomic datasets, the authors investigate the role of cross talk between DC and NK cells in the pathogenesis of CL.

There are several concerns: the results are based on only two datasets.

Several of the implicated genes are also specific to other cell types and just specific to NK and/or DC.

NKp30 is specific to NK cells; are its ligands expressed on DC.

CCR7 is negatively correlated with CL size; what it has to do with NK-DC cross talk?

The discussion should discuss the molecule pairs (ligands vs receptors) which are reciprocally expressed on DC/NK cells and are implicated in the cross talk.

On the whole, I do not think that the data signify the role of DC/NK cross talk in the pathogenesis of CL.

By the way, what is the section ‘0. How to Use This Template’?
